# Protein-Based Predictive Biomarkers to Personalize Neoadjuvant Therapy for Bladder Cancer—A Systematic Review of the Current Status

**DOI:** 10.3390/ijms25189899

**Published:** 2024-09-13

**Authors:** Stacy Bedore, Joshua van der Eerden, Faizan Boghani, Saloni J. Patel, Samer Yassin, Karina Aguilar, Vinata B. Lokeshwar

**Affiliations:** Department of Biochemistry and Molecular Biology, Medical College of Georgia, Augusta University, 1410 Laney Walker Blvd., Augusta, GA 30912, USA; sbedore@augusta.edu (S.B.); jvandereerden@augusta.edu (J.v.d.E.); spatel16@augusta.edu (S.J.P.); samer.yassin@nm.org (S.Y.); kaaguilar@augusta.edu (K.A.)

**Keywords:** bladder cancer, neoadjuvant therapy, lineage plasticity, protein-based biomarkers, systematic review, prognostic markers, tumor heterogeneity

## Abstract

The clinical outcome of patients with muscle-invasive bladder cancer (MIBC) is poor despite the approval of neoadjuvant chemotherapy or immunotherapy to improve overall survival after cystectomy. MIBC subtypes, immune, transcriptome, metabolomic signatures, and mutation burden have the potential to predict treatment response but none have been incorporated into clinical practice, as tumor heterogeneity and lineage plasticity influence their efficacy. Using the PRISMA statement, we conducted a systematic review of the literature, involving 135 studies published within the last five years, to identify studies reporting on the prognostic value of protein-based biomarkers for response to neoadjuvant therapy in patients with MIBC. The studies were grouped based on biomarkers related to molecular subtypes, cancer stem cell, actin-cytoskeleton, epithelial–mesenchymal transition, apoptosis, and tumor-infiltrating immune cells. These studies show the potential of protein-based biomarkers, especially in the spatial context, to reduce the influence of tumor heterogeneity on a biomarker’s prognostic capability. Nevertheless, currently, there is little consensus on the methodology, reagents, and the scoring systems to allow reliable assessment of the biomarkers of interest. Furthermore, the small sample size of several studies necessitates the validation of potential prognostic biomarkers in larger multicenter cohorts before their use for individualizing neoadjuvant therapy regimens for patients with MIBC.

## 1. Introduction

Urothelial cell carcinoma is a major contributor to global cancer morbidity and mortality and is one of the costliest cancers in clinical management [1,2]. Due to healthcare systems unique to each country, healthcare access and reimbursement issues also affect how patients with bladder cancer are managed globally [3,4]. Arising from urothelial transformation, two clinical characteristics of bladder cancer are unique, frequent recurrence and divergent clinical behaviors of low-grade and high-grade tumors, notwithstanding the heterogeneity of mixed pathologies. Low-grade tumors are mostly confined to the urothelial layers (stage Ta). Although high-grade tumors could be nonmuscle-invasive at the time of diagnosis (stage Ta or T1–lamina propria invasion), nearly two-thirds are diagnosed as muscle-invasive bladder cancer (MIBC; stage T2 or more). Stage T3 tumors have invaded the perivesical fat tissue, and stage T4 tumors have invaded the adjacent organs [5]. Carcinoma in situ is a high-grade tumor confined to the urothelium. The standard of care for low-grade tumors is transurethral resection of a bladder tumor (TURBT) followed by surveillance for recurrence. High-grade non-MIBC is treated with TURBT followed by an intravesical treatment, most often BCG (Bacillus Calmette–Guérin). In cases of BCG refractory high-grade non-MIBC, BCG-related adverse events, and/or BCG shortage, chemotherapy instillations (e.g., gemcitabine-based combinations) are increasingly being used to reduce the risk of recurrence [6,7].

The preferred first-line therapy for muscle-invasive bladder cancer (MIBC) is cystectomy with pelvic lymphadenectomy [8]. Neoadjuvant chemotherapy (NAC) offers the possibility of complete tumor resection due to down-staging and the treatment of micrometastases. NAC plus radical cystectomy has shown superior overall survival compared to cystectomy alone [9,10,11]. Recent comparisons across tumor stages show that, compared to cystectomy alone, NAC combined with radical cystectomy improves overall survival and pathological complete response in patients with T3-4aN0M0 bladder cancer, although this benefit is not observed in patients with T2N0M0 [12]. During the COVID-19 epidemic, the EAU guidelines Rapid Reaction Group recommended omitting NAC for patients with T2N0M0 disease [13]. Platin-based chemotherapy regimens are preferred with methotrexate vinblastine doxorubicin cisplatin (MVAC), gemcitabine plus cisplatin, and, more recently, dose-dense MVAC as the common regimens for NAC. Among these, dose-dense MVAC has shown superiority as it reduces the interval between NAC and cystectomy; however, the regimen has significantly higher grade 3–4 toxicities. Therefore, the careful selection of patients with biomarkers that predict the response to a chemotherapy regimen or identify patients with cisplatin ineligibility is necessary when choosing NAC followed by cystectomy versus cystectomy alone [14]. Nevertheless, there is paucity of studies reporting biomarkers for predicting response to intravesical chemotherapy. Immunotherapy based on immune checkpoint (PD-1/PD-L1) inhibitors (pembrolizumab, atezolizumab) is approved for treating non-MIBC; however, the response is variable, and may not always correlate with PD-L1 expression, thus necessitating the need for biomarkers to identify patients who may benefit from immunotherapy [15,16,17].

Platinum-based (cisplatin-based) chemotherapy is the recommended first-line standard therapy for patients with advanced/metastatic bladder cancer. Nearly 50% of patients unfit to receive cisplatin-based chemotherapy may benefit from carboplatin-based therapy. Patients positive for PD-L1 but unfit to receive cisplatin may receive immune checkpoint inhibitor (atezolizumab or pembrolizumab) therapy [18]. Maintenance immunotherapy with avelumab is a standard of care for all patients with stable disease following first-line platinum-based chemotherapy [16,18]. Curiously, however, similar to non-MIBC, the response to immune checkpoint inhibitors does not always correlate with PD-L1 expression, which, again, stresses the clinical need for prognostic biomarkers for predicting the response to immunotherapy [19].

Molecular classification of bladder tumors has subdivided MIBC into six or more molecular subtypes, which appeared to hold promise for predicting response to both chemotherapy and immunotherapy as well as patient outcomes [20,21]. However, molecular subtyping in the present form may not be ready for clinical practice because of tumor heterogeneity and lineage plasticity limiting their clinical utility [5,22]. As has been realized for breast cancer, functional drivers and not molecular subtypes assigned by transcriptome profiling, per se, may be more useful in predicting a bladder tumor’s clinical behavior. 

As discussed above, immunotherapy is indicated based on PD-L1 expression. Atezolizumab and pembrolizumab show prolonged progression-free survival in patients with locally advanced or metastatic bladder cancer as a first-line treatment in cisplatin-ineligible patients or platin-noneligible patients [23,24]. High PD-L1 expression is generally associated with a higher percentage of pathological complete response with immunotherapy in the neoadjuvant and adjuvant settings [25,26,27]. However, in a Phase III trial, the addition of pembrolizumab to first-line platinum-based chemotherapy did not significantly improve efficacy, and the recommendation was that it should not be widely adopted for the treatment of advanced urothelial carcinoma [28]. In addition to PD-L1 expression, tumor mutation burden, defined as the number of somatic nonsynonymous mutations per mega base, could potentially predict response to immunotherapy; higher tumor mutation burden is associated with improved therapy response [15,16,24]. Although in the early stages of research, bladder tumors plausibly have distinct metabolome signatures, and it remains to be determined if the metabolome signatures correlate with clinical outcome, including prediction of treatment response [29,30]. While the current emphasis is on multiomics to predict clinical outcome, immunohistochemistry for functional drivers of bladder cancer performed on whole tumor specimens is one potential way to counter heterogeneity and lineage plasticity. In this systematic review, we examined the available evidence for prognostic protein-based biomarkers for response to systemic therapy in MIBC within the last five years.

## 2. Methods

### Search Strategy

A systematic search of the literature was conducted according to the Preferred Reporting Items for Systematic Reviews and Meta-analyses (PRISMA) statement [31]. The PubMed database was searched between 1 January 2019 and 31 May 2024 to identify studies reporting on the prognostic value of protein-based biomarkers for response to systemic therapy in patients with MIBC. The search strings “(bladder cancer) AND (neoadjuvant treatment)) AND (marker)” and “(bladder cancer) AND (neoadjuvant treatment)) AND (molecular subtype)” were utilized. The review process was conducted in two parts. The first was an initial screening of the title and abstracts to identify eligible publications; next, a full-text reading of selected articles for final inclusion and data extraction was performed. We excluded any non-English language articles, review articles, editorials, case reports, and duplicates between the two search strings. The following variables were extracted from each selected article: sample size, single vs. multi-institution design, tumor grade, pathological TNM staging (tumor stage, lymph node invasion, metastasis), NAC regimen, definitions of clinical outcome and treatment failure, follow-up duration, biomarkers studied and marker detection method, the reported sensitivity and specificity of the marker, and if the marker was found to be an independent prognostic marker.

## 3. Results

### 3.1. Literature Review

Based on the criteria for Cochrane systematic reviews “to identify, appraise and synthesize all the empirical evidence that meets prespecified eligibility criteria to answer a specific research question”, we developed a comprehensive search strategy to answer a research question: Do protein-based markers described in the literature accurately predict response to NAC? [32]. We searched articles indexed in PubMed using predefined search criteria. The strategy for selecting the studies for review is shown in Figure 1. With the exclusion criteria of filtering out publications before the last 5 years, the initial online search identified 155 publications as our initial sample size, with two search strings: 1. “Bladder cancer” and “neoadjuvant therapy” and “marker”; and 2. Bladder cancer” and “neoadjuvant therapy” and “molecular subtype”. With the elimination of duplicates from the two strings and based on the predefined inclusion criteria of original research work, we eliminated review articles, case reports, and editorials, as well as articles without a description of the biomarkers, those only discussing diagnostic markers or nonprotein-based biomarkers, and those with inadequate clinical data including follow-up. Fourteen studies met all the predefined criteria and, thus, were included in this review. Table 1 provides an overview of the selected articles, including the biomarkers studied and the significant findings.

### 3.2. Predicting Neoadjuvant Chemotherapy Response

The availability of immunotherapy and combination chemotherapies in the NAC setting provides options for choosing a regimen that will result in a better clinical outcome while minimizing adverse events. However, a treatment that is not effective poses a significant challenge because the disease may progress while the patient is receiving the treatment. Biomarkers that predict response to NAC and add specificity for a particular NAC could personalize the NAC regimen(s) and minimize the risk for treatment failure.

### 3.3. Using IHC to Characterize Subtypes and Their Response to NAC

Over the past decade, molecular subtypes of bladder cancer have been identified through transcriptome analysis. While there are multiple classification schemes that have been utilized, the two most prominent subtypes that have been identified are the basal and luminal subtypes, as well as a p53-WT subtype. Luminal bladder tumors are similar in their gene expression pattern to the more differentiated intermediate and superficial layers of normal urothelium. Basal cancers recapitulate the gene expression signature of the less differentiated basal layer of the urothelium [47,48]. Previously, luminal subtypes had been shown to have the best overall survival, with basal subtypes exhibiting the best response to NAC [49]. Recent studies have aimed to identify biomarkers via immunohistochemistry to determine these subtypes and provide an accessible and more cost-effective alternative to transcriptome analysis [33,34,35,36,37]. 

The urothelial basal cell layer is characterized as urothelial stem cells that express cytokeratins 5 and 14 (KRT 5, 14). The basal cell layer of the urothelium give rise to intermediate cells, which then differentiate into the umbrella (superficial) cells; KRT20 is predominantly expressed in luminal cells. KRT5/6 and KRT14 have been identified as markers for the basal subtype and KRT20 and GATA3 for the luminal subtype [50]. Razzaghdoust et al. used KRT5/6(+)/KRT20(-) as a signature to define the basal subtype. In their study, the expression of this signature in bladder tumors was significantly associated with complete response to NAC (*p* = 0.037) [36]. Of note, GATA3 and KRT14 were not considered in this study, as they were present in either most or almost no tumor samples, respectively. Furthermore, no significant difference was found in terms of overall survival between any IHC-based subtype (*p* = 0.721).

Helal et al. considered the p53 subtype in addition to the luminal and basal subtypes [33]. KRT5/6 was used as a biomarker for basal subtype and GATA3 for luminal subtype [51]. Basal tumors had a significantly better response to NAC compared to both luminal and p53-WT tumors, with 45% showing a complete response. In addition to the markers for these subtypes, Helal et al. also assessed the prognostic significance of human epidermal growth factor 2 (HER2) and epidermal growth factor receptor (EGFR) expression in bladder tumors. Both growth factor receptors have been implicated in other types of cancers, including bladder cancer [52,53,54]. While the predictive capabilities of these markers were not studied, their expression was significantly elevated across bladder cancer subtypes (*p* < 0.001 for both). HER2 was positive in 9/12 (75%) of the luminal subtype tumors, with only 3 (25%) basal subtype and no p53 subtype tumors staining positive. EGFR was positive in all basal subtype tumors, but only six (22%) luminal and two (18.2%) p53-type tumors were positive for EGFR. However, no multivariate analysis was performed to determine the prognostic potential of EGFR and HER2 in predicting response to NAC [33]. 

In contrast to the two studies above, Sjodahl et al. found that the basal/squamous subtype showed inferior pathological complete response to NAC compared to luminal subtypes. Of note, the LundTax 13-marker IHC panel was used to determine the molecular subtypes, and luminal subtype was further divided into multiple genomically unstable and urothelial-like subtypes [55]. Pryma et al. sought to utilize Uroplakin II (UPII) as a marker for luminal subtype compared to basal [35]. UPII is a marker of urothelial cell differentiation, and it protects the urothelium from harmful substances [56]. There was no significant difference in the overall or cancer-specific survival between high and low UPII groups, and the response to NAC was not studied. Finally, Morselli et al. assessed the predictive capability of a four-marker IHC panel of KRT20, KRT5/6, GATA3, and CD44 [34]. The KRT20+, KRT 5/6-, GATA3+, and CD44- pattern showed a significantly better response to NAC compared to KRT20-, KRT5/6+, GATA3-, and CD44+ and all other patterns (*p* = 0.021, 0.013, respectively). When individually assessed, only KRT20-positive staining was significantly associated with pathological complete response to NAC (*p* = 0.013). However, a small sample size limited the statistical power of this study. These studies show a lack of consensus among subtype-related markers in predicting response to NAC. This could be because of the heterogeneity in the expression of these markers in MIBC specimens, the markers used in each study to define a subtype, and further subtyping of the markers without a unifying system or consensus [22].

Tumor heterogeneity stems from the fact that tumors represent a heterogeneous population of cells with different mutational burden, transcriptomes, and protein and metabolomic signatures. Therefore, a single tumor is likely to have more than one molecular subtype within the same tumor environment, let alone in different parts of the tumor (e.g., hypoxic center versus invading tumor edge), thus limiting the utilization of molecular subtyping in predicting response to NAC. While this phenomenon is more established in other cancers such as lung and prostate, studies are limited in bladder cancer [57,58]. One report has demonstrated differences in molecular subtypes between bladder tumors and corresponding lymph node metastasis, but studies of intratumoral heterogeneity have had mixed results [59,60,61]. 

Schallenberg et al. sought to characterize the heterogeneity of molecular subtypes within MIBC bladder tumors by examining the tumor center and invasive front with IHC markers of molecular subtypes including FGFR3, CCND1, RB1, CDKN2A, KRT5, KRT14, FOXA1, GATA3, TUBB2B, EPCAM, CDH1, and vimentin [38]. They also investigated the relationship between intratumoral heterogeneity and disease-specific survival. Ultimately, intratumoral subtype heterogeneity was found in 12.5% of cases between the tumor center and invasive front and 24.6% of cases within any tumor location, but no specific molecular subtypes were associated with intratumoral heterogeneity. Only 9.3% of all cases of intratumoral heterogeneity had more than two molecular subtypes. Intratumoral heterogeneity was also significantly more prevalent in lower stage tumors (≤pT2), compared to advanced stages (pT3 and pT4), present in 38.7% and 21.9% of tumors, respectively (*p* = 0.046). The basal subtype was more prevalent in advanced (pT4) tumors vs. earlier stage (pT2) tumors, present in 26.2% and 11.5% of tumors, respectively (*p* = 0.049). It is possible that among the several tumor cell clones that are initially present in a clinically localized tumor (lower stage: non-MIBC), invasive tumor cells that survive in different levels of hypoxia predominate in advanced tumors. Nevertheless, disease-specific survival was not associated with intratumoral heterogeneity, whether present in tumor center and invasive front (*p* = 0.379), within any location (*p* = 0.125), or between molecular subtypes (*p* = 0.995). A limitation to this study is that although all patients were reported to have undergone radical cystectomy, the study design did not specify if the patients also received NAC, and what regimen, complicating disease-specific survival predictions. 

### 3.4. Cancer Stem Cell Markers as Predictors of NAC Response

Stem cells serve vital roles in normal physiology, helping to maintain homeostasis and cell repair by their ability to differentiate into multiple cell types. Cancer stem cells promote tumor growth and metastasis, upregulate signaling pathways that increase multidrug resistance, and, therefore, contribute to poorer outcomes and chemoresistance [62,63]. Blomqvist et al. assessed three potential biomarkers for cancer stem cells, viz., aldehyde dehydrogenase 1 (ALDH1), sex determining region Y-box 2 (SOX2), and stage-specific embryonic antigen-4 (SSEA-4) [39]. ALDH1 has been implicated as a biomarker for cancer stem cells and regulates pluripotency via the retinoic acid pathway [64,65]. Among patients with bladder cancer, high ALDH1 expression has been associated with both poor outcomes and worse clinicopathological features [66,67,68]. SOX2 also regulates pluripotency, maintains stem cell-like characteristics [69,70,71], and has been associated with poor survival in multiple cancers, including bladder cancer [72,73,74,75]. SSEA-4 is a ganglioside expressed in embryonic stem cells and is a specific cell surface marker for pluripotent stem cells. High expression of SSEA-4 in tumor cells has been associated with poor survival among patients with lung, breast, brain, and prostate cancers [72,73,74,75]. SSEA-4 likely promotes increased tumor growth and chemoresistance in multiple types of cancer and is a possible treatment target in glioblastoma multiforme [76]. Contrary to the reported association of these three stem cell markers with poor clinical outcome, only the negative/weak staining of SOX2 in the cytoplasm was associated with lymphovascular invasion and nonorgan confined disease. Furthermore, none of the three markers were significantly associated with disease progression, response to NAC, or cancer-specific survival. It should be noted, however, that in this study, 61% had non-MIBC, which could have impacted the conclusion of this study.

### 3.5. Epithelial–Mesenchymal Transition (EMT) Markers as Predictors of NAC Response

EMT of tumor cells is a process that is associated with both tumor invasion and metastasis. One component of this transition is the “cadherin switch” [77,78]. Prior to EMT, E-cadherin facilitates contact inhibition by forming tight junctions between cells. As EMT occurs, E-cadherin expression decreases, and expression of N-cadherin and vimentin (a marker for mesenchymal cells) increases. Hensley et al. utilized both cadherin types and vimentin as biomarkers for EMT transition, as well as zeb-2 and β-catenin [40]. β-catenin is known to function as part of the WNT-signaling pathway but is also associated with calcium-dependent cell adhesion, forming protein complexes with E-cadherin that facilitates EMT. Accumulation of β-catenin in the nucleus also induces transcription of both c-Myc and cyclin D1, thus contributing to tumor growth and progression [79]. 

Hensley et al. studied the expression of these markers in case-matched pre-NAC TURBT and post-NAC cystectomy specimens. The response was measured as organ confined disease and extravesicular disease following NAC and cystectomy. The extravesicular disease cohort had increased expression of N-cadherin, vimentin, and β-catenin (*p* = 0.004, 0.028, and 0.019, respectively) in pre-NAC specimens. However, in multivariate analysis, only increased N-cadherin expression significantly correlated with the extravesicular disease (*p* = 0.027). N-cadherin was also a significant prognostic indicator for disease-specific mortality (*p* = 0.016). Furthermore, N-cadherin, vimentin, and zeb-1 predicted a complete pathologic response (*p* = 0.044, 0.013, and 0.030, respectively) [40].

### 3.6. Apoptosis-Related Markers and Response to NAC

Platinum-based chemotherapies, typically included in NAC for MIBC, are known to induce apoptosis by cross-linking DNA which interferes with DNA replication and repair mechanisms, thus setting into motion the apoptotic cascade. This mechanism makes biomarkers of apoptosis attractive as potential predictive markers of response to NAC treatment. Expression of Bcl-2, an antiapoptotic effector of the Bcl-2 family, has been extensively investigated as a prognostic indicator or predictor of chemotherapy response in multiple cancer types [80,81,82]. However, reports of predictive or prognostic correlations of Bcl-2 expression in MIBC have been mixed [83,84]. Multiple recent studies have investigated Bcl-2 expression by immunohistochemistry as a predictor of response to NAC; however, none found Bcl-2 to be predictive. Hensley et al. reported no difference in Bcl-2 expression between those who did and did not progress on NAC (*p* = 0.390) [40]. Likewise, in two independent studies, Türker et al found no significant association between Bcl-2 expression status and response to NAC (*p* = 0.38) [41,42]. Bcl-2 also did not have any independent prognostic value in predicting 5-year overall survival in a competing risk model.

Türker et al. further investigated other biomarkers associated with apoptosis, Emmprin and Survivin, as well as CCTα [42]. Emmprin (extracellular matrix metalloproteinase inducer), is a glycoprotein expressed on the surface of tumor cells. It functions to stimulate the production of matrix metalloproteinases by adjacent stromal cells [85]. Survivin is an intracellular regulator of cell division and survival and is upregulated in many cancers [86]. Immunohistochemical expression of Emmprin and Survivin was previously found to be predictive of response to cisplatin-based chemotherapy in advanced bladder cancer [87]. CCTα (choline phosphate cytidylyltransferase-α) expression is elevated in cancer and was previously studied as a predictor of NAC response in MIBC [88,89]. In their 2021 study, Türker et al. found only CCTα expression to be predictive of response to NAC, as measured by overall survival, with negative CCTα expression being associated with better outcomes in the NAC treated group compared to the untreated group (*p* = 0.009). In a combined biomarker model (Bcl-2, Emmprin, Survivin, and CCTα), only patients with negative CCTα and Emmprin expressions benefited from NAC compared to patients positive for either marker individually (*p* < 0.001) [42].

Hensley et al. used two different assays to measure rates of apoptosis in bladder cancer specimens obtained by TURBT [40]. As would be expected from the action of chemotherapy regimens, they found statistically significant higher rates of apoptosis among samples from patients who responded to NAC (complete pathologic response ≤ypT2N0) compared to those who progressed on NAC (≥ypT3 or N+; *p* = 0.002) after multiple comparisons adjustment, indicating higher apoptosis rates are predictive of response to NAC. In combination, these studies indicate that biomarkers of apoptosis could be promising predictors of response to NAC.

Taniyama et al. evaluated the prognostic potential of Schlafen 11 (SLFN11) to predict response to platinum-based adjuvant therapy or NAC [43]. Schlafen 11 (SLFN11) is a putative DNA/RNA helicase that induces irreversible replication block and is, therefore, a sensitizer for platinum-based chemotherapy. Consistent with its function as a DNA/RNA helicase, in bladder tumor specimens, SLFN11 expression was nuclear and increased expression associated with better clinical outcome following NAC or adjuvant chemotherapy. Furthermore, while SLFN11 expression did not correlate with any clinicopathologic parameters, higher expression was an independent predictor of better prognosis. SLFN11 expression was also associated with a luminal marker GATA3. Although the size of the cohort in this study was limited, these clinical findings corroborate the preclinical studies which showed SFN11 knockdown induced resistance to cisplatin and other platin derivatives in multiple bladder cancer cell lines [43].

### 3.7. Actin/Cytoskeleton Organization and Its Potential to Predict Response to NAC

The cytoskeleton plays an important role in migratory and infiltrative capabilities of cancer cells. Cofilin, a member of the actin-depolymerizing factor family, promotes the polymerization and depolymerization of actin filaments. Regulation of this activity is accomplished through phosphorylation and dephosphorylation of the N-terminal Serine3. Cofilin has previously been shown to affect cell migration in response to TGF-β in prostate cancer [90]. Elevated cofilin expression is also associated with poor prognosis in some cancers [91]. Hensley et al. found a correlation between cofilin/phosphorylated-cofilin (P-cofilin] expression and high-stage/high-grade bladder cancer [92]. These authors subsequently sought to determine if cofilin, P-cofilin, or α-tubulin levels were associated with response to NAC in patients with muscle-invasive bladder cancer [40]. Tumor specimens from patients who progressed on NAC (≥ypT3 or N+) had higher expression of cofilin (*p* = 0.148), P-cofilin (*p* = 0.036), and α-tubulin (*p* = 0.007) compared to those who responded to NAC (ypT0). However, higher α-tubulin expression was the only marker significantly associated with poor NAC response in this subset of patients after multiple comparisons adjustment across all three actin/cytoskeleton markers (*p* = 0.037). It is possible that actin-cytoskeletal markers may be associated with response to NAC, and, among these, α-tubulin expression could be further evaluated.

### 3.8. Tumor-Infiltrating Immune Cells for Predicting Response to NAC

The immune system’s role in response to cancer development and progression has long been investigated [93]. In MIBC, the presence of tumor-infiltrating immune cells, specifically CD8^+^ T cells, has been shown to be a prognostic indicator for predicting clinical outcomes [94]. Tumor-infiltrating Forkhead box P3^+^ (FoxP3) T regulatory cells have, likewise, been associated with improved survival in MIBC [95]. Wahlin et al., thus, sought to study if the expression of various tumor-infiltrating immune cells was predictive of response to NAC in the setting of MIBC [44]. They examined the densities of CD8^+^ and FoxP3^+^ T cells as well as CD20^+^ B cells prior to and at cystectomy in patients who received or did not receive NAC. Bladder tumor specimens from paired TURBT, cystectomy, and lymph nodes were examined. CD20^+^ B cells were included in this study because while B cells have not been investigated in the setting of bladder cancer, B cell infiltration in tumor tissues has been noted as a prognostic indicator in other cancers [96]. Nevertheless, here, researchers found no differences in the densities of any of the immune cell types in TURBT specimens (prior to NAC) between noncomplete responders (T stage ≥ 1) and complete responders (T stage 0 or Ta/CIS). Furthermore, biomarker expression was not significantly different between TURBT and cystectomy specimens of noncomplete responders, although, in congruence with previous reports, the study found an association of prolonged time to recurrence with high CD8^+^ T cell infiltration in TURBT specimens and for all cell types in cystectomy specimens by Kaplan–Meier and Cox regression analyses.

The authors also studied the expression of the immune checkpoint modulator Programed Death 1 (PD-1) and its ligand (PD-L1) as predictive markers of response to NAC. PD-1 is expressed in activated T cells and engages with PD-L1 expressed on tumor cells. Engagement transduces a signal that inhibits T-cell proliferation, cytokine production, and cytolytic function [97]. As discussed above, PD-1/PD-L1 immunotherapy is approved for metastatic bladder cancer treatment. However, previous studies are inconsistent in showing a correlation between response to PD-1/PD-L1 immunotherapy and its expression by IHC [15,16]. Wahlin et al. reported an independent association of PD-1 and PD-L1 expression in immune cells with prolonged time to recurrence, while PD-L1 expression in tumor cells did not show such a correlation [44]. As with the immune cell types studied, PD-1 and PD-L1 expression was not predictive of response to NAC, nor was there a difference in expression between TURBT and cystectomy specimens in NAC nonresponders. This study’s findings indicate that while PD-1 and PD-L1 expression may have prognostic value for MIBC, they do not appear to be predictors of NAC response. 

Although a PubMed search using the terms “bladder cancer”, “neoadjuvant therapy”, and “biomarkers” resulted in articles published on clinical trials in this area, two recent clinical trials LCCC1520 (NCT02690558) and BLASST-1 (NCT03294304) deserve consideration for two reasons. First, both of these trials incorporated immune checkpoint inhibitors into NAC regimens for MIBC, resulting in pathologic response rates ranging from 36 to 66%, with improved recurrence-free survival. Second, the specimens from these trials before therapy were used to analyze the ability of biomarkers to predict response. The LCCC1520 was a Phase II study to evaluate the efficacy and toxicity of gemcitabine and split-dose cisplatin plus pembrolizumab for patients with clinical T2-4aN0/XM0 disease in an NAC setting [98]. Similarly, the BLASST-1 trial was a multicenter Phase II study to determine the safety and efficacy of nivolumab when given in combination with cisplatin and gemcitabine in the NAC setting for patients with T2-4aN0M0 disease. Beckabir et al. analyzed 18 responders and 18 nonresponders of NAC (gemcitabine and cisplatin) plus one of the immune checkpoint inhibitors pembrolizumab (LCCC1520) or nivolumab (BLASST-1) by proteomics digital special profiling (DSP)-based marker expression to develop a predictive model of neoadjuvant chemoimmunotherapy (NACI) response [45]. The expression of 52 protein markers associated with various immune function/profiling were examined by DSP. NACI responders were found to have significantly higher expression of markers associated with the upregulation of immune checkpoints and antigen presentation including PD-L1, CTLA4, and HLA-DR, while nonresponders had significantly higher expression of markers of increased vascularity, Treg cells, and immunosuppressive stroma (CD34, FAP-alpha, CD127, and fibronectin). The authors examined the expression of these 52 protein markers in different locations within the tumor microenvironment to determine if the markers’ differential expression in tumor-enriched vs. immune-enriched areas (or their interfaces) was predictive of NACI response. They found that increased Ki67 staining in immune-enriched tumor locations (2-fold increase in responders; *p* ≤ 0.001) and HER2 staining in tumor predominant locations (3.5-fold increase in responders; *p* ≤ 0.001) were associated with NAC response. The authors used the results of DSP with integrated differences in location within the tumor microenvironment to develop an elastic net (EN) response model for predicting NACI response. While DSP has the potential to predict NACI, the small sample size was the major limitation of the study. 

Myint et al. investigated another immune cell marker, CD47 [46]. CD47 is a transmembrane receptor expressed on tumor cells that promotes immune invasion by inhibiting the phagocytosis of tumor cells by macrophages. CD47 expression is elevated in several cancers, including bladder cancer. In bladder cancer, CD47 expression has been reported to be higher in T1 tumors [99]. In their study, Myint et al. examined CD47 expression in matched TURBT and cystectomy specimens in patients who did and did not receive NAC. However, in both univariate and multivariate analyses, the researchers found no associations between CD47 levels and NAC response. Although CD47 expression was lower in post-NAC specimens, the results were statistically insignificant. CD47 levels also did not appear to be an independent predictor of disease recurrence or overall survival by Kaplan–Meier curves and log-rank test [46]. Therefore, CD47 expression is less likely to be predictive of NAC response. 

## 4. Discussion and Future Directions

A review of the literature from the last five years on prognostic protein-based biomarkers for response to systemic therapy in MIBC brings to light several issues for study appraisal and future clinical applications. Comparisons between studies are hindered by a lack of uniformity in protocols. Currently, there are no universal protocols for diagnostic workup, tissue procurement for biomarker analysis, IHC staining, or evaluating a biomarker’s potential to predict response to therapy in patients’ MIBC. This may account for some of the discrepancies between studies reviewed here, such as Blomqvist et al., and the prior literature. In their study, Blomqvist et al. examined ALDH1, SOX2, and SSEA-4, and found none to be associated with a poor response to NAC [39]. This is not consistent with previous studies that reported on the predictive capability of such cancer stem cell markers, including ALDH1 and SOX2 [75,100]. The differences in the findings could be attributed in part to the variability in cancer stem cells in different parts of the tumor. For example, hypoxic tumor centers would have higher expression of the cancer stem cell markers than the periphery, but this may not necessarily predict clinical outcome. Blomqvist et al. evaluated the prognostic significance of nuclear and cytoplasmic SOX2 expression, but neither reached statistical significance [39]. The discrepancies in the prognostic potential of these markers among different studies could also be because of small sample size or the use of different antibodies and scoring systems that are used to evaluate the staining, thus emphasizing the need for uniformity among studies when evaluating the prognostic potential of biomarkers.

Apoptotic indices were assessed by Hensley et al. and by Türker et al [40,41]. Neither of these studies found BCL2 to be predictive of NAC response. Furthermore, the latter study found that BCL2 positivity, and, therefore, low apoptosis activity, was associated with better survival, which is contradictory to the tumor-promoting role of BCL2. This finding could be due to multiple factors, including antibody specificity or the fact that staining intensity was not considered, and that tumors were marked as BCL2-positive only if >10% of the sample stained positive. A subsequent publication by Türker et al. found that CCTα was predictive of NAC response [42]. However, no staining images were shown, impeding assessment of the staining pattern and the grading method. Furthermore, both nuclear and plasma membrane staining patterns were evaluated for markers which are known to have specific localizations. Moreover, it has been previously shown that CCTα expression does not correlate with response to NAC in patients with bladder cancer [89]. 

Hensley et al. also found that N-cadherin, vimentin, and zeb-1, markers of EMT, were significantly associated with complete pathological response. This study included matched TURBT and cystectomy specimens and a small number of lymph node metastases. These findings should be confirmed in a larger study. Intratumor heterogeneity in MIBC may also represent a challenge to the utilization of tissue microarray for IHC biomarker predictability [38,101]. Multiple genetic populations exist within a tumor which may express different biomarkers at different levels and may respond differently to therapy. Intratumor heterogeneity has been studied in the context of molecular subtypes of bladder cancer and has a logical extension to IHC staining techniques as well. While one prior study found low intratumor heterogeneity in bladder cancer tumors by IHC, further studies are needed with predictive biomarkers [61].

Although molecular subtyping of MIBC is based on molecular transcriptome data, the major subtypes, luminal and basal, have also been evaluated at the protein level. The most studied markers are KRT5/6 and KRT14 for basal tumors and GATA3 and KRT20 for luminal tumors. Assessing luminal vs. basal subtypes with IHC as predictors of NAC response revealed mixed results. Razzaghdoust et al. found that KRT5/6(+)/KRT20(−) tumors, as a marker for basal subtype, had significantly better complete response to NAC [36]. Similarly, Helal et al. found that KRT5/6-positive tumors had superior response to NAC. Of note, P53-WT tumors were found to be chemoresistant, but the study was limited because mutation status was determined only by scoring diffuse vs. scattered staining patterns and no sequencing data were presented. Therefore, it is unclear if the tumors truly expressed wild-type P53 or also harbored mixed populations with mutated P53 [33]. While these studies showed increased NAC response in basal subtypes, Sjodahl et al. found that basal subtype was associated with worse NAC response. Nevertheless, in this study, the 13-antibody LundTax marker panel was used, with IHC subtype assigned based on a four-marker score for each subtype [37]. Morselli et al. found that KRT20+, KRT 5/6-, GATA3+, and CD44- tumors, denoting a luminal subtype, were associated with better NAC response, with only KRT20+ showing predictive capabilities individually. However, IHC data were shown only for KRT5/6 and KRT20, and the sample size was insufficient to assess the prognostic value of this four-marker signature [34]. Using publicly available datasets and clinical cohorts, we previously showed that the molecular subtypes could not predict clinical outcome and response to treatment [22]. Pryma et al. studied Uroplakin II as a marker for luminal subtype but did not evaluate at NAC response [35]. Because these subtypes have limited predictive capabilities, the need for a correlated biomarker is unclear, especially when the subtypes themselves are already defined by biomarkers [22]. By analyzing multiple cohorts, including the TCGA, our previous study demonstrated, for the first time, that molecular subtypes have limited prognostic ability to predict response to NAC and clinical outcome. This finding was recently confirmed by Lerner et al., who concluded that a “Consensus Classifier”, based in part on the TCGA, MDA, and COXEN classifiers, had only a modest predictive power to predict pathologic downstaging following NAC, and the subtypes did not have prognostic capability to predict progression-free or overall survival [22,102]. This systematic review of protein-based markers shows that like the genome- and transcriptome-based markers, IHC-based markers are not currently ready for translation into the clinical arena, and will require a concerted multicenter effort to choose the potential markers and develop a consensus both on the methodology for assaying them and the rigor with which to analyze their efficacy.

## 5. Conclusions

Overall, these studies provide insight into how immunohistochemistry could be employed as a tool to lessen the challenges posed by tissue heterogeneity and lineage plasticity that may be more significant when evaluating prognostic markers by other methods. However, heterogeneity can still become an obstacle when utilizing tissue microarrays and should be considered. Standardized scoring systems for analyzing staining are key for better and more reliable assessment of the biomarkers of interest. Finally, many of the studies mentioned had small sample sizes, and expanding these studies to larger cohorts could allow for more statistical power in validating new potential predictive biomarkers.

## Figures and Tables

**Figure 1 ijms-25-09899-f001:**
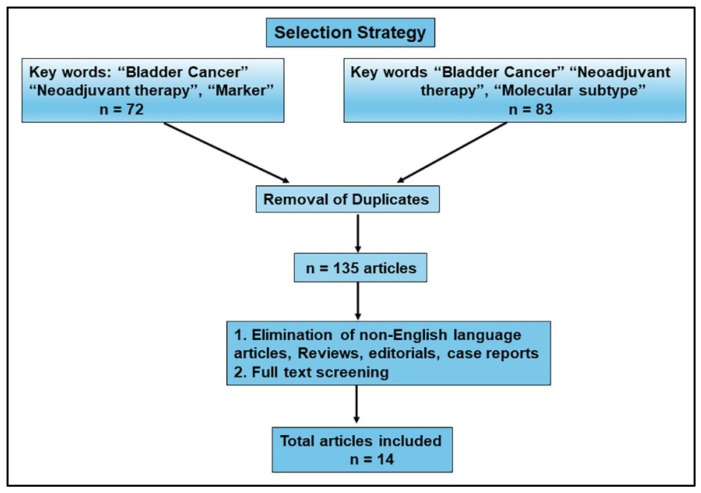
**Flow diagram.** Results from literature review. With the exclusion criteria of publications from the last 5 years, search strings resulted in 72 and 83 hits, for an initial sample size of 155. After the removal of duplicates, 135 articles were further analyzed, as shown in the flow diagram. Based on the inclusion parameters, and after full-text appraisal, 14 publications were selected for inclusion in this review.

**Table 1 ijms-25-09899-t001:** Summary of studies included in comprehensive literature review.

PMID & Reference	^a^ Markers	Sample Size	^d^ Study Type	Tumor Grade	T-Stage	NAC Regimen	^i^ Clinical Outcome	^j^ Statistical Methods	Predictive Biomarkers (*p*-Values)
36737799 [33]	**KRT5**, **KRT6**, GATA3, p53, HER2, EGFR	60	Single	HG	T2–T4	^f^ Gem+Cis; Cis ineligible Gem+Carbo	CR; OS; DFS	KM; Cox	NAC response correlates with KRT5/6 (+) (basal subtype (*p* = 0.045)
34494414 [34]	KRT5, KRT6, **KRT20**, CD44, GATA3	16	Single	Not provided	Ta–T4a	Gem + Cis	CR; PR	Descriptive	KRT20 (*p* = 0.013) small study
35612672 [35]	UPII	80	Multi	^e^ Not provided	T2–T4	Not provided	OS; CSS	KM	Not significant Prediction of luminal subtype–cut-off 50% -> 90% accuracy
33943049 [36]	**KRT5**, **KRT6**, KRT20	63	Multi	LG (2); HG (61)	LG: Ta; HG tumors: T2–T4	Gem+Cis; cis ineligible—Gem+Carbo	CR; OS	LR; KM; Cox	KRT5/6 (+)/KRT20 (−) combined marker (*p* = 0.037)
34782206 [37]	13-Marker LundTax Panel	148	Multi	Gx, G2, G3	T2–T4	Cis-based	CR; PR; NR	KM; Cox	Basal/squamous subtype: Poor response; osteopontin–subtype dependent response prediction
37076398 [38]	^c^ 12 markers	208	Single	Not provided	T2–T4	Not provided	DFS	KM; LR; Chi-sq; FE	None significant
34211078 [39]	ALDH1, SOX2, SSEA-4	269	Multi	LG (1); HG (68); Unknown (5)	Ta, Tis, T1 = 22 T2–T4	Gem+Cis (90% pts); Gem+Carbo (10% pts)	CR; PR	KM; Chi-sq	None significant
31326313 [40]	E-cadherin, **N-cadherin**, **vimentin**, **zeb-1**, β-catenin, cofilin, phospho cofilin **α-tubulin**	90	Multi	Not provided	T2	MVAC or Gem+Cis	OC vs. EV; CR	LR; KM; WSR; HM	NAC response prediction: N-cadherin (*p* = 0.027); *α*-tubulin (*p* = 0.037)
30806186 [41]	Bcl-2	247	Multi	Not provided	T2–T4	MVAC	OS	LR; KM; Cox	Not significant
33766713 [42]	Emmprin, Survivin, Bcl-2, **CCTα**	639	Multi	Not provided	T2–T4	MVAC	OS	KM; Cox	CCTα (*p* = 0.009); CCTα (-)/Emmprin (-) better survival
34808009 [43]	**SLFN11**	120	Multi	LG (6); HG (44)	T2–T4	Carbo or Cis based	OS	KM; Cox	SLFN11 (*p* = 0.018)
31646091 [44]	CD8, FoxP3, CD20, PD-1, PD-L1	135	Single	HG	T2–T4	^g^ MVAC; plus adjuvant chemotherapy	CR; PR	KM; Cox	None significant
38092611 [45]	^b^ proteomic digital spatial profiling–52 markers	36	Multi	Not provided	T2–T4	^h^ Gem+Cis + Immunotherapy	OS; CR; PR	KM, FE, WSR, DL	Intratumoral heterogeneity of DSP markers; certain DSP-based markers (Ki67; HER2) in a spatial context potentially predict response
37373873 [46]	CD47	87	Single	HG	T2–T4	Cis-based	CR; PR	KM; Cox	Not significant

a: Bold if found to be predictive of NAC response. b: DGAPDH, Histone H3, Ms IgG1, Ms IgG2a, Rb IgG, S6, CD127, CD25, CD27, CD40, CD44, CD80, ICOS, PD-L2, Beta-2-microglobulin, CD11c, CD20, CD3, CD4, CD45, CD56, CD68, CD8, CTLA4, Fibronectin, GZMB, HLA-DR, Ki-67, panCK, PD-1, PD-L1, SMA, CD14, CD163, CD34, CD45RO, CD66b, FAP-alpha, FOXP3, 4-1BB, ARG1, B7-H3, GITR, IDO1, LAG3, OX40L, STING, Tim-3, VISTA, Bcl-2, EpCAM, ER-alpha, HER2, MART1, NY-ESO-1, PR, PTEN, S100B. c: FGFR3, CCND1, RB1, CDKN2A, KRT5, KRT14, FOXA1, GATA3, TUBB2B, EPCAM, CDH1, and vimentin. d: Study type: multi, multi-institutional; single, single institutional. e: Not provided; based on tumor stage, the grade most likely should be high grade. f: Adjuvant radiotherapy if T3, T4 disease. g: 12 patients received adjuvant chemo for advanced disease. h: Immunotherapy: pembrolizumab or nivolumab. i: CR: complete response (pT0); PR: tumor downstaging; NR: nonresponder; OS: overall survival; DFS: disease-specific survival; CSS: cancer-specific survival; OC: organ confided; EV: extravesical. j: KM: Kaplan–Meier; LR: logistical regression; WSR: Wilcoxon signed rank test; HM: Holm’s method for multiple comparisons; FE: Fisher’s exact test; DL: DeLong’s tests of area under the receiver curve.

## Data Availability

Not applicable.

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
