# Peer review of "Protein-Based Predictive Biomarkers to Personalize Neoadjuvant Therapy for Bladder Cancer—A Systematic Review of the Current Status"

_ijms, 2024, doi:10.3390/ijms25189899_

Round 1
Reviewer 1 Report
Comments and Suggestions for Authors
The present review manuscript, “Protein-based Predictive Biomarkers to Personalize Neoadjuvant Therapy for Bladder Cancer—A Systematic Review of the Current Status,” is a systematically written review. The authors have elaborated on the potential of protein-based biomarkers as prognostic markers to predict the outcome of neoadjuvant therapy regimens for patients with muscle-invasive bladder cancer (MIBC). The authors discussed identifying 135 articles, which they further sorted out, and ended up with 14 research articles briefly explained in this review systemically. The authors have also provided pitfalls and a necessary outlook on expanding some studies to larger cohorts for better results and outcomes to predict the potential of potential predictive biomarkers for NAC regimens for bladder cancer. This manuscript, with a few minor corrections or editing, could significantly impact the field of oncology and personalized medicine in bladder cancer, enhancing its clarity and relevance for the reader. Below are some suggestions that the authors should consider:
1. The authors should briefly explain the bladder cancer stages for better clarification.
2. The authors should briefly provide an explanation of KRT5/6(+)/KRT20(-) as they have mentioned other potential biomarkers.
3. The authors should provide brief insights into some of the ongoing clinical trials and their outcomes if that comes under the scope of their work. Though they mentioned LCCC1520 (NCT02690558) and BLASST-1 (NCT03294304), perhaps some elaboration and explanation would be better for understanding the importance of finding potential predictors of NACI response.
4. The authors have abruptly ended the discussion, which seems incomplete or missing some text. The authors should check that and add the required text to complete the discussion and future directions section.
Reviewer 2 Report
Comments and Suggestions for Authors
Title - this is not a systematic review, but a narrative one - title is imprecise - Major
Abstract - concise review of the essence of the study - No remarks
Introduction -
"In cases of Bacillus Calmette-Guerin (BCG) refractory high-grade non-MIBC, BCG-related adverse events, and/or BCG shortage, chemotherapy instillations (e.g., Gemcitabine-based combinations) are increasingly being used to reduce the risk of recurrence (12,13). Nevertheless, there is paucity of studies reporting biomarkers for predicting response to intravesical chemotherapy" - not related to study topic - predictive biomarkers for NAC - Major
Results - in-depth analysis of the contemporary literature from the past 5 years - No remarks
Discussion, future directions and conclusion - precise description of the discrepancies, limitations and contradictory results of the studies - No remarks
Reviewer 3 Report
Comments and Suggestions for Authors
The manuscript is a systematic review of the studies reporting potential prognostic biomarkers for the use of neoadjuvant chemotherapy (NAC) and immunotherapy in patients with muscle invasive bladder cancer (MIBC) within the last five years.
The introduction points out the clinical need for standard prognostic biomarkers in order to identify patients eligible for NAC and/or immunotherapy treatments, minimize the risk of adverse effects and better predict the tumor’s outcome and response to the treatment.
Methods are well written. Results section reviews and compares the chosen studies. However, the results need to be addressed to improve the understanding of the manuscript.
Overall, the manuscript is well presented and articulated and can be utilized as a starting point to improve the research on new prognostic biomarkers for the treatment of MIBC.
Minor revisions
· 2. Results- Literature review
The process of paper selection and the flowchart are not clear.
In the text of the paragraph, it would be clearer to include that the initial search based on the selected key words brought to a total of 155 papers and after a removal of 20 duplicates the total number was 135. The flowchart seems to indicate the removal of 135 duplicated papers.
· Table 1:
The authors have to revise table 1. The letters b, g and h in superscripts reported in the legend are not included in the table.
Round 2
Reviewer 2 Report
Comments and Suggestions for Authors
Authors have sufficiently addressed the reviewer`s concerns
will accept the explanation for the systematic review with reserves - the essence of systematic review is mathematical compilation of results form different studies for better level of evidence